# Printing Drugs onto Nails for Effective Treatment of Onychomycosis

**DOI:** 10.3390/pharmaceutics14020448

**Published:** 2022-02-19

**Authors:** Thomas D. Pollard, Margherita Bonetti, Adam Day, Simon Gaisford, Mine Orlu, Abdul W. Basit, Sudaxshina Murdan, Alvaro Goyanes

**Affiliations:** 1Department of Pharmaceutics, UCL School of Pharmacy, University College London, 29-39 Brunswick Square, London WC1N 1AX, UK; thomas.pollard.19@ucl.ac.uk (T.D.P.); margherita.bonetti01@universitadipavia.it (M.B.); adam.day@ucl.ac.uk (A.D.); s.gaisford@ucl.ac.uk (S.G.); m.orlu@ucl.ac.uk (M.O.); a.basit@ucl.ac.uk (A.W.B.); 2Fabrx Ltd., 7B North Lane, Canterbury CT2 7EB, UK; 3I+D Farm Group (GI-1645), Departamento de Farmacología, Farmacia y Tecnología Farmacéutica, Universidade de Santiago de Compostela, 15782 Santiago de Compostela, Spain

**Keywords:** point-of-care, onychomycosis, inkjet printing, personalized healthcare, antifungal treatment, pharmaceutical 2D printing, desktop printing, printed drug products, precision pharmaceuticals

## Abstract

Inkjet printing (IJP) is an emerging technology for the precision dosing of medicines. We report, for the first time, the printing of the antifungal drug terbinafine hydrochloride directly onto nails for the treatment of onychomycosis. A commercial cosmetic nail printer was modified by removing the ink from the cartridge and replacing it with an in-house prepared drug-loaded ink. The drug-loaded ink was designed so that it was comparable to the commercial ink for key printability properties. Linear drug dosing was shown by changing the lightness of the colour selected for printing (R^2^ = 0.977) and by printing multiple times (R^2^ = 0.989). The drug loads were measured for heart (271 µg), world (205 µg) and football (133 µg) shapes. A disc diffusion assay against *Trpytophan rubrum* showed inhibition of fungal growth with printed-on discs. In vitro testing with human nails showed substantial inhibition with printed-on nails. Hence, this is the first study to demonstrate the ability of a nail printer for drug delivery, thereby confirming its potential for onychomycosis treatment.

## 1. Introduction

Inkjet printing (IJP) is an important technology for individualised healthcare [1], with applications ranging from biosensor production [2,3,4] to cell printing [5,6]. IJP is the deposition of small volumes of liquid from a cartridge. The cartridge expels the liquid using either thermal or piezoelectric actuation with precise control over the position. This control makes IJP an attractive technology for pharmaceutical research [7] and has showcased easier drug loading [8], increased data storage using QR codes [9], high loading capacities [10], improved dissolution rates [11,12], compatibility with continuous manufacturing [13] and medical device loading [14,15]. IJP has already been illustrated with various drugs [16,17,18,19] as well as probiotics [20]. Additionally, different antimicrobials have been printed onto gauze [21] and microneedles [22,23,24,25].

IJP has revolutionised cosmetic nail design. Whereas previously, skilled individuals had to carefully decorate nails by hand, IJP allows users to print intricate designs easily. Commercial printers for inkjet printing onto nails are now available for home and salon use. The user inserts their finger into the printer, selects the design they wish to print using an app on a wirelessly connected mobile device or inbuilt screen, and then the printer prints this pattern onto their nail. The pattern can be a design from the app’s catalogue, or users can select their own design, giving the user a truly individualised pattern. These nail printers boast ease of operation, rapid printing times (approximately 10 s for one nail) and the ability to print detailed artistic designs.

The greatest benefit from nail IJP could be for onychomycosis therapy. Onychomycosis is the fungal infection of a nail and is the most common nail disease [26], affecting 10% of the general population and up to 50% of people aged over 70 years [27]. The most common cause of onychomycosis is *Trichophyton rubrum* [28], a dermatophyte. Symptoms of onychomycosis include nail discolouration, thickening, brittleness and even pain [29]. Effective treatment improves a patient’s quality of life [30,31], prevents the infection from progressing to more severe states [32], and limits infection transmission. The most effective antifungal drug for the treatment of onychomycosis caused by dermatophytes is terbinafine. Terbinafine is a synthetic allylamine derivate [33] whose mode of action is through fungal squalene epoxidase inhibition [34]. It is moderately soluble in ethanol (45 mg/mL) but has low solubility in water (3 mg/mL) [35], and shows thermal stability up to 175 °C [36].

The market for onychomycosis treatment is growing [37], and new innovations, such as laser devices [38], are emerging. Clinics are now offering this nail treatment, despite limited and contradictory evidence of clinical efficacy and the potential for adverse effects such as pain and bleeding [29,39,40,41]. Oral treatment is fairly effective (60–76%) [42], but this has a long treatment period and increased side effects [43]. An effective topical treatment would avoid systemic side effects through local and targeted action. However, the efficacies of currently licensed medicines can be as low as 6% [44]. Hence, improved topical treatment is essential. Current research is ongoing into different drug carriers and topical formulations [45,46,47,48,49], including nail patches [50], UV-curable gels [51,52,53], lacquers [54,55,56], films [57,58] and solutions [59,60,61]. However, many of these applications do not allow for the user to precisely control the dose, or to easily apply multiple drugs.

The aim of this paper was to evaluate the use of a two-dimensional nail printer for administering medicines against onychomycosis. A commercially available cosmetic nail printer was modified by removing the existing ink and replacing it with an in-house prepared drug-loaded ink, with terbinafine hydrochloride. Terbinafine is the most effective antifungal drug against onychomycosis caused by dermatophytes. The ink was tested for its printability, and the dispensed drug was measured. The adapted nail printer was then used to print onto antimicrobial discs and human nail clippings, and the antifungal effectiveness of the prints was tested using disc diffusion assays against *T. rubrum*.

## 2. Materials and Methods

### 2.1. Materials

The ethanol used in this study was puriss grade (Sigma Aldrich, Gillingham, UK) and the water used was ultrapure grade (Type I), from a Triple Red Water Purification System (Avidity Science, Long Crendon, UK). The red food colourant used was Kroma Kolors Red (Kopykake Enterprise Inc., Torrance, CA, USA). The drug used, shown in Figure 1a, was pharmaceutical secondary standard terbinafine hydrochloride (HCl) (Sigma Aldrich, Gillingham, UK). For the HPLC mobile phase, the chemicals used were ≥99.9% Acetonitrile (Sigma Aldrich, Gillingham, UK), ≥99% Trimethyl amine (Sigma Aldrich, Gillingham, UK) and 85–90% Phosphoric acid (Thermo Fisher Scientific Inc., Loughborough, UK).

The fungus used was *Trichophyton rubrum* strain ATCC 28188 (Thermo Fisher Scientific Inc., Loughborough, UK). The agar plates were prepared from Sabouraud Dextrose Agar (Sigma Aldrich, Gillingham, UK). In addition, McFarland turbidity standard 0.5 (Thermo Fisher Scientific Inc., Loughborough, UK) and Whatman antibiotic assay discs, 6 mm (Fisher Scientific, Loughborough, UK), were used.

Human nail clippings were collected from volunteers following informed consent and ethics approval (4359/004) from University College London.

### 2.2. Preparation of the Terbinafine HCl Solution ‘Inks’

A 35 mg/mL terbinafine HCl-loaded ink was prepared by adding 350 mg of the drug to a 10 mL volumetric flask followed by 7.5 mL of ethanol. The flask was then vortexed and ultrasonicated to fully dissolve the terbinafine HCl. Then, 1.76 mL of water was measured out using a PIPETMAN L P5000L Gilson Pipette (Gilson Inc., Middleton, WI, USA) and added into the volumetric flask. The mixture was vortexed and made up to 10 mL with ethanol. The final ethanol to water ratio was 82.4:17.6 *v*/*v*, or 80:20 *w*/*w*. Two drops of a red food colourant were added and mixed so that the printed solution could be visualized. The mass of drug added was reduced for less concentrated solutions, as appropriate.

### 2.3. Characterization of the Commercial and In-House Prepared Drug Inks

Various techniques were used to characterise the inks and nozzle to ensure that the inks were printable. All measurements were done in triplicate.

#### 2.3.1. Density

The density of the cosmetic and prepared drug-loaded inks was measured by placing the solution on a Precisa 180A weighing balance (Precisa Balances Ltd., Livingston, UK), taring the balance to zero, removing a set volume of solution using a PIPETMAN L P200L Gilson Pipette (Gilson Inc., Middleton, WI, USA) and recording the change in mass. The density of the inks was then calculated by dividing the change in mass by the volume of solution removed.

#### 2.3.2. Viscosity

Viscosity measurements were made using AMVn Automated Micro Viscometer (Anton Paar Ltd., St Albans, UK), controlled by VisioLab Windows Software (Anton Paar Ltd., St Albans, UK). A 1.6 mm or 1.8 mm diameter glass capillary was used with a 1.5 mm ball. The 60 × 4 program was used throughout.

#### 2.3.3. Surface Tension

Surface tension was measured using a Kibron Delta-8 microtensiometer (Kibron Inc., Helsinki, Finland) in conjunction with a 96-well Dyneplate (Kibron Inc., Helsinki, Finland). 50 µL of solution was added into the well, with Type 1 water used as the calibration solution.

#### 2.3.4. Scanning Electron Microscope (SEM)

The cartridge nozzle was removed from the body of the cartridge, splutter-coated with gold and mounted onto a disc mount. Measurements of the printing nozzle diameter were made using a JSM-840A Scanning Microscope (JEOL GmbH, Freising, Germany). Measurements were made at 10.00 kV, with magnifications of 110, 400 and 6000×. SEM images were analysed using the Digitizer v4.5 (Ankit Rohatgi, Bay Area, CA, USA) online image analysis software.

### 2.4. Calculating Suitable Ink Properties

In order to produce drug-loaded inks that were suitable for printing, the properties of the inks needed to match those of the commercial cosmetic inks. Printability is predicted using the *Z* value, calculated from Equation (1) [62,63,64];
(1)$$Z=\frac{{\left(\alpha \;\rho \;\gamma \right)}^{\frac{1}{2}}}{\mu }$$where *α* is the diameter of the printing orifice (µm), *ρ* is the density of the solution (g cm^−3^), *γ* is the surface tension (mN·m^−1^) and *μ* is the dynamic viscosity (mPa s). A stable droplet at the printing nozzle is formed when the *Z* value is between 4 and 14 [63,64].

### 2.5. High Performance Liquid Chromatography

The concentration of the drug in the liquid ink was determined using High-Performance Liquid Chromatography-UV (HPLC-UV), using a Hewlett Packard 1260 Series HPLC system (Agilent Technologies, Stockport, UK). The method used was adapted from a previously reported procedure [50]. The stationary phase was a Luna C18 Column, 150 × 4.6 µm (Agilent Technologies, Stockport, UK), and the mobile phase was a combination of aqueous solution and Acetonitrile at a ratio of 65:35 *v*/*v*. The aqueous solution was 0.012 M Trimethylamine and 0.020 M Phosphoric Acid in Type I water. The flow rate was set to 1 mL/min, with a column temperature of 40 °C, an injection volume of 20 µL and a UV-wavelength of 224 nm. The drug peak showed an elution time of approximately 10.3 min. A calibration curve for terbinafine HCl in solution was obtained between 1 and 30 µg/mL, with a correlation of R^2^ = 0.9993. The HPLC calibration solutions were also used to estimate the stability of the drug in solution. The solutions were stored in sealed 1.5 mL amber glass vials (Fisher Scientific, Loughborough, UK).

To measure the drug load in a printed film, the specified shape was printed onto an Academy 22 × 22 mm glass coverslip, 0.16–0.18 mm thick (Camlab Ltd., Over, UK). This was then washed into a 5 mL volumetric flask using drug-free ethanol and water solution. The washing was filtered through a 0.45 µm filter (Millipore Ltd., Tullagreen, Ireland), and the concentration determined by HPLC. By multiplying the concentration found by 5 mL, the mass of the drug printed could be calculated.

### 2.6. Inkjet Printer and Printing Process

The inkjet printer used in this study was the O2Nails V11 Printer (Guangzhou Taiji Electronic Co. Ltd., Guangzhou, China), a two-dimensional inkjet printer designed for printing onto human nails, loaded with the standard SM10 Special Inkjet Cartridge (Guangzhou Taiji Electronic Co. Ltd., Guangzhou, China) (Figure 1b). The ink cartridge is comprised of 3 separate ink cartridges, containing yellow, magenta and cyan inks. The composition of these commercial inks is proprietary and thus not disclosed. The inkjet printer was controlled by the O2Nails app (Figure 1c) (Guangzhou Taiji Electronic Co, China) via WiFi, allowing the user to control the system using an iPad Mini 2 smart tablet (Apple Inc., Cupertino, CA, USA) with iOS 12.4.5 software. The inkjet printer also includes a small camera. The camera was used to align the printing surface and visualise the prints during printing. The app includes automatic nail detection and allows users to upload their own designs and images for printing.

The ink cartridges (Figure 1d) were emptied and thoroughly cleaned of the commercial inks before use. The green cover was removed, and the clear plastic covering was drilled through and removed along with the sponges and inks. The empty cartridge was cleaned by ultrasonication with ethanol inside. The cartridge was placed on a beaker during ultrasonication to avoid water damage. The ultrasonication was repeated with fresh ethanol until it remained colourless, indicating no ink remained. A cleaned cartridge is shown in Figure 1e.

In order to print, the drug-loaded ink was pipetted into the yellow compartment of the cartridge, and the cartridge was covered with parafilm and left for at least 30 min to allow the formulation to reach the printing nozzle. The bottom of the nozzle was wiped with an ethanol damped paper towel to remove any excess before the cartridge was placed into the inkjet printer. The green cover was placed on top of the parafilm. This was required to trigger the cartridge detection switch in the printer to allow it to print. The printer was ‘primed’ by printing three large rectangles to remove any remaining ink from previous printing and ensure the ink had fully reached the nozzle. Printing was controlled from the connected tablet, with the shape and size of the printed area determined by the user. The cartridge nozzle was wiped regularly with an ethanol damped paper towel to remove any excess ink. This prevented drug crystallization from forming and blocking the nozzle.

Both glass and polystyrene plastic were tested as a surface to print on. These were printed onto, and the print was washed off into a 5 mL volumetric flask and analysed via HPLC. Each measurement was made in triplicate. The method for extracting the print from the surface was also tested by both soaking the surface in an ethanol:water solution as well as washing it into the volumetric flask.

The printer was tested by printing a series of black and white shapes. Further, some of these were tested for their drug load using HPLC. Each shape tested for its drug load was printed with 2 passes, with each shape printed in triplicate.

### 2.7. Characterisation of the Prints with X-ray Diffraction (XRD)

XRD was conducted using a Rigaku MiniFlex 600 (Rigaku, The Woodlands, TX, USA) with a Cu K α X-ray source of wavelength λ = 1.5418 Å. The range of angles used was 2θ = 3–60°, with a step size of 0.02° and a speed of 5°/min. The intensity was 15 mA, with an applied voltage of 40 kV.

Tests were done on the sample pan, pure drug powder and printed solution. For the printed solution, an XRD pan was directly printed onto 20 times, with the nozzle wiped every 3 prints. It was then allowed to dry fully before being loaded into the machine and tested.

### 2.8. Controlling the Dose of Drug Printed

The printed drug dose was controlled by both varying the lightness of the image printed and by printing multiple times. As the ink was loaded into the yellow ink compartment, the lightness of the colour yellow was changed to alter the drug load. In both experiments, each value was measured in triplicate and the area of the printed shape was kept constant throughout. The order in which the different shades or passes were printed was randomised to reduce any systematic errors.

The lightness of the printed image was changed through 7 different shades of yellow between 0 and 88%, from the CMYK colour model. To print multiple times, printing was conducted on the same glass slide between 1–6 times, to give 6 different values.

### 2.9. In Vitro Antifungal Efficacy

#### 2.9.1. Preparation of Media

Sabouraud Dextrose Agar (SDA) was used to prepare agar plates for the growth of *T. rubrum*. Using an analytical balance, 26 g of SDA powder was weighed and placed into a 500 mL Duran bottle with 400 mL distilled water. The bottle was tightly closed, shaken well to obtain an even suspension and then autoclaved at 121 °C for 2 h. This was cooled and poured into sterile Petri dishes in a microbiological safety cabinet. Once the Petri dishes had solidified at room temperature, they were stored at 4 °C until used.

#### 2.9.2. Preparation of Fungus Inoculum

*T. rubrum* isolate was used in this study. Using sterile inoculating loops, a swab of *T. rubrum* was taken and streaked onto an SDA agar plate. The plate was incubated for 7 days at 32 °C to allow *T. rubrum* growth.

The inoculum for the efficacy assays was prepared immediately before the experiment as follows: 5–10 mL of sterile saline solution (0.85% NaCl in distilled water) was pipetted onto the fungal colonies on the plate. The surface of the agar gel was then scraped using a sterile loop to give a suspension of dermatophyte hyphae and conidia. A 5 mL plastic syringe was then used to collect the fungal suspension, which was subsequently filtered through a 110 mm diameter Whatman’s filter paper, Grade 40 (Cytiva, Marlborough, MA, USA) in an autoclaved stainless-steel filter holder in order to remove all dermatophyte hyphae and collect the micro-conidia in the filtrate. The conidial suspension was vortexed for 15 s.

Following this, the turbidity of the suspension was measured using a Libra S12 UV/Vis spectrometer (Biochrom, Cambridge, UK), at a wavelength of 530 nm. The target absorbance was 0.15–0.17 AU, similar to that given by a McFarland standard. The conidial suspension was adjusted with more saline or conidial suspension until the target AU was achieved, equivalent to a concentration of 4 × 10^3^ to 5 × 10^4^ colony forming units (CFU)/mL. This suspension was used without further dilution in the disc diffusion assays.

#### 2.9.3. Disc Diffusion Assays

This method is based on the determination of the zone of inhibition (ZOI) of antifungal drugs (Figure 2a,b) when impregnated onto a disc and then placed onto agar gel inoculated with fungi. Prior to the experiment, all discs were placed in a small Duran bottle and autoclaved at 121 °C for 2 h.

The drug solution was printed onto a disc using the nail printer, with the discs held in place using an in-house produced plastic holder. The printed shape was a circle of diameter 4 mm. Care was taken to ensure the edge of the discs were not printed on, thus ensuring that the drug must permeate through the disc and onto the agar gel to inhibit fungal growth. Once the printing was finished, the printed-on discs were allowed to dry and then placed onto an agar plate, as shown in Figure 2a. Each Petri dish was divided into quadrants, with 3 quadrants used for the drug-loaded discs and 1 for the control, drug-free disc. The control disc was printed on with the same number of passes using an equivalent, drug-free solution. Then, 100 µL of fungal inoculum was loaded onto the SDA Petri dish. The inoculum was spread using a sterile spreader to evenly distribute the inoculum on the Petri dish. Once the prepared discs were thoroughly dry, sterile tweezers were used to transfer the discs onto the Petri dish, as shown in Figure 2a, and the Petri dish was then incubated at 32 °C for 7 days. The zones of inhibition, indicated in Figure 2b, were inspected visually.

#### 2.9.4. Antifungal Susceptibility Testing Using Printed Nail Clippings

The efficacy of terbinafine HCl printed onto nail clippings was assessed using the disc diffusion assay described in Section 2.9.3, except the drug solution was printed onto nail clippings, rather than on antimicrobial discs. The printed nail clippings acted as ‘discs’ in the disc diffusion assay. The nails were flattened between two metal plates, held together by G-clamps, and left overnight. They were then sterilised by autoclaving at 120 °C for 20 min. Once cooled, the nail clippings (Figure 2c) were cut in half (Figure 2d), placed into an in-house prepared nail holder (Figure 2e), which included a small window. The window allowed for printing onto a specific area to ensure that the nail edges were print-free. The nail clipping holder was inserted into the nail printer and printed on the desired number of times. Once the ink had dried (Figure 2f), the nail was removed from the holder and placed onto the *T. rubrum*-inoculated Petri dish and incubated at 37 °C for 7 days. The nail was checked to make sure that the printed ink did not go over the edge of the nail (Figure 2g). The zones of inhibition from this were also inspected visually.

## 3. Results and Discussion

### 3.1. Printer Optimisation

Density, surface tension and viscosity measurements of the commercial inks are shown in Table 1. Drug loaded inks were then prepared to closely match the properties of the commercial inks. Ethanol and water were chosen as the ink solution, as reported values for the density, surface tension and viscosity of water–ethanol mixtures [65] were found to be appropriate for printing.

The ratio of ethanol:water used was chosen to give high drug loadings to enable higher efficacies. Previous data has shown terbinafine HCl is thermally stable up to 175 °C [36], and so is well suited to thermal inkjet printing, where heat is used to dispense the droplet [66]. Terbinafine HCl is more soluble in ethanol (45 mg/mL) than water (3 mg/mL). Hence, choosing a higher ethanol:water ratio gave higher terbinafine HCl concentrations in the solution while still retaining printability. The ratio chosen for ethanol to water was 80:20 *w*/*w*. This ratio has previously been used in creating a terbinafine HCl loaded-lacquer [67]. A small amount of food colouring was added to allow for visualisation of the prints.

The stability of the terbinafine HCl solutions was tested by looking at the change in the HPLC area under curve (AUC) measurements for the calibration solutions stored in ambient conditions after a period of 66 days. The average absolute percentage difference was 3.7%, indicating that the drug remained highly stable in solution over this time frame. The majority of samples showed a decrease in AUC, although a few showed a small increase which may be due to solvent evaporation.

Using SEM (Figure 3a–c), the nozzle diameter of the ink cartridge was measured as 21.0 ± 2.2 µm. Density, surface tension and viscosity measurements of the ethanol–water solution, along with the addition of 1 mg/mL terbinafine HCl and food colouring, are shown in Table 1.

The results from this show that the values of the key parameters of these water and ethanol-based solutions are similar to the values of the commercial inks, suggesting that the solutions will be printable. The values obtained are also in good agreement with literature values [65]. It is also noteworthy that the addition of 1 mg/mL terbinafine HCl and food colouring did not significantly alter the properties of the solution.

The drug-loaded ink was added into a cleaned cartridge, covered in parafilm and the green lid was added on top to trigger the cartridge detection switch on the printer. This solution was found to print effectively, and so was used for further studies. The prints from this were also detectable using HPLC, indicating that the drug molecules were not degraded during printing.

XRD was performed to investigate if the printing process changed the phase of the drug. The results from XRD are shown in Figure 3d. The background data from the glass slide gives no sharp peaks but does have some broad trends, indicating that it is amorphous. The distinctive peaks seen in the drug powder sample, especially at 2θ = 5.9°, indicate that the pure drug is crystalline.

The dried printed sample also shows a distinctive peak at 2θ = 5.9° but none of the other peaks. Hence, terbinafine HCl is partially crystalline after printing. This is not expected to affect the antimicrobial activity of the drug.

The surface for printing and the extraction method of the print were both tested. No significant statistical differences were found between printing on glass or polystyrene (two-tailed unpaired Student’s *t*-test, *p* = 0.49), or between washing into the volumetric flask versus soaking the printed-on surface in solution (two-tailed unpaired Student’s *t*-test, *p* = 0.38). It can thus be concluded that either surface can be used for printing onto and either method of extracting the print can be chosen. Hence, glass slides were selected for printing onto, and the drug in the print was extracted by washing it with ethanol:water solution into a volumetric flask.

### 3.2. Personalised Dosing

The printer was tested for its ability to print different shapes. A series of images (Figure 4a) were printed onto paper and compared with the original images. Three of the images; the heart, world and football shapes, were also printed onto glass slides, and their drug loads were measured by HPLC, shown in Figure 4b.

The printed shape results show that the printed shapes were all reproducible to a high degree of accuracy. This is promising for personalised dosing, as being able to print thin lines, such as those seen in the dog or basketball shape, will allow for precise control over the drug dose. Additionally, the printer is capable of printing solid shapes, as seen with the heart and UCL logo, suggesting that the printer can also print high drug doses.

High drug loads were found for the football (133 ± 16 µg), world (205 ± 7 µg) and heart (271 ± 3 µg) shapes. It is not surprising that the heart had the highest drug load as, from visual inspection, this has the highest amount of coloured area, and so more ink is required to print the shape. In each case, the prints showed that high levels of drugs could be printed. This demonstrates the potential of the printer for use in a clinical setting as it is able to accurately reproduce shapes and give a predictable pattern of drug loading.

For the nail printer to be a viable topical drug delivery tool, the dose of drug dispensed must be controlled. One method investigated for controlling the dose printed was to change the lightness of the printed object. The ink was loaded into the yellow ink compartment, and so printing different lightnesses of yellow should change the amount of ink the inkjet printer dispenses, and thus alter the drug dose. The results from this, shown in Figure 5a, are highly linear (R^2^ = 0.977). This indicates that changing the lightness of the printed object is an effective method for controlling the dose.

In addition to controlling the drug load by changing the lightness of the colour, the potential for controlling the drug dose by printing the same shape multiple times was tested. The results from this are shown in Figure 5b. As expected, this is highly linear, with a value of R^2^ = 0.989. This is anticipated—the printer should print the same amount of drug regardless of the number of previous prints, and so the drug load per print should be constant. Hence, printing more times gives a higher drug load.

Both of these methods demonstrate that it is possible to control the dose printed with a high degree of linearity. This is important in any practical setting since the amount of drug applied will directly impact the patient’s outcome.

### 3.3. In Vitro Antifungal Activity

In order for the nail printer to be an effective method for treating onychomycosis, it must be able to demonstrate effective antifungal activity. This was tested using both discs and human nail samples.

#### 3.3.1. Disc Diffusion Assay

The first antifungal assay used the antibiotic assay discs. Once they were printed onto and left to dry, the discs were placed with the printed side face-up on the agar plates in order to determine the ability of the drug to diffuse through the disc and inhibit the growth of the fungi. The results from this are shown for both 1 and 2 print passes with 29 µg/mL terbinafine HCl solution (Figure 6a–f).

As can be seen, in both cases there is a clear inhibition of the fungi in the test quadrant compared to the control quadrants. Clearly, the printed drug is capable of inhibiting the growth of *T. rubrum*, even at very low doses. The size of the inhibition zone is similar to that reported in previous studies [68]. This is hugely promising, and the dose can be increased by a factor of 1000× to give much stronger inhibition. For this study, low doses were adequate.

Hence, it is apparent that this delivery system is effective in printing drugs onto discs, and that the printed drug can permeate through the disc and into the agar plate. The drug has retained its ability to inhibit the growth of the fungi despite being printed, and so the printing process did not alter the drug significantly. This was also seen in the XRD data, where the printed drug retained the main crystalline peak of the drug powder. The drug being unaffected by printing is important to allow printing to become a treatment modality.

In Figure 6a–c, fungal inhibition is seen to be fairly strong with distinctive zones of inhibition. However, on some plates, there is little inhibition (Figure 6a) or no inhibition (Figure 6d). Since the printing process was not always consistent (Figure 5a,b), the irregular drug load from printing may have given inconsistent drug loading into the agar gel, and thus the varying extents of fungal inhibition. Additionally, inconsistent diffusion of the printed drug through the antimicrobial disk and permeation into the agar gel may be a factor in the range of fungal inhibition seen in Figure 6a–f.

#### 3.3.2. Nail Diffusion Assay

Once the effectiveness of the printing method had been established for assay discs, the nail printer was tested on human nails using a similar method. These nails were tested against *T. rubrum* with a terbinafine HCl concentration of 5 mg/mL, with the results shown in Figure 6g–i. The drug load of these was 5.3 ± 0.6 µg, whilst the controls showed no drug content.

Figure 6g–i shows a clear inhibition of the fungus for the test quadrants, but not the control quadrants of the agar plates; each test nail is surrounded by clear agar, whereas all the control nails are covered by *T. rubrum*. Thus, it is clear that this method is suitable for inhibiting the growth of *T. rubrum* by drug printing onto the nail plate, and thus it could be used to treat onychomycosis. This is especially promising, as the drug must pass through the nail and into the agar plate in sufficient quantities. It should be noted that there is some inconsistency with the size of inhibition zones, which is likely due to the irregular nature of nails. As with the disc diffusion assay, the drug dose here can also be further increased to give even stronger inhibition of the fungi should this be necessary. One point of note is that some of the nails in Figure 6g–i showed a small amount of microbial growth on them. This does not appear to be *T. rubrum*, due to the nature and colour of the growth.

### 3.4. General Discussion

Herein, we have demonstrated that the nail printer has the potential for personalised dosing as an effective treatment for onychomycosis. The system outlined here boasts a number of advantages over the existing literature. Unlike previous inkjet printing systems which printed onto different materials [21,22,23,24,25,69], this system allows for users to apply the drug directly onto the nail at the point of care. It also poses an advantage as it requires no additional steps [70]. However, in order for this to be translated into a real-world setting, there are a number of steps that need to be taken.

In the future, this technology could be used in a podiatric or nail salon setting. For example, following a fungal infection diagnosis, the healthcare practitioner would select the design and program to directly print the best drug or drug combination onto the infected nail. The drugs could be included in a cosmetic nail print to improve the appearance of the diseased nail. Given the printer cartridge contains 3 ink wells, the printer could contain different antifungal drugs for printing depending on the onychomycosis-causing agent. Drug combinations could also be printed to give enhanced antifungal activities [71]. Since there is a need to improve drug permeation into the nail [46,72], chemicals that enhance drug permeation could also be added to some of the cartridges. Following the drug print, an additional occlusive layer could also be printed to improve drug efficacy, as occlusion is known to enhance nail plate hydration and thus increase drug penetration into the nail plate [73]. Occlusion would also enable germination of drug-resistant fungal spores into drug-susceptible fungal elements to further enhance treatment.

For this appliance to be most effective, a custom-designed nail printing system should be developed. This system could allow for different cartridges to be loaded with different concentrations of the drug, or even different drugs, drug combinations or chemical ungual enhancers [74]. Hence, the drug loading could be tightly controlled, multi-drug therapies could be achieved and topical treatment could be enhanced. Furthermore, a custom system could be optimised to achieve the highest drug loading possible and thus increase the efficacy of the treatment through this. In addition, a custom-designed system could be combined with digital software imaging for identifying the areas most in need of treatment and measuring the effectiveness of the treatment over time. It would also be possible to match the properties of the printer and the ink to reduce the number of steps required by, for example, removing the need to wipe the nozzle between passes. Finally, this system could include additional elements that would further benefit the patient, such as a cartridge for printing the previously mentioned cosmetic and/or occlusive layers.

## 4. Conclusions

An existing nail art printer was successfully modified to allow for the printing of a drug-loaded solution. An in-house prepared drug solution was adjusted to match the properties of the original commercial ink in order to retain printability. This system was then shown to be able to control the dose of the drug by changing the colour lightness of the image selected for printing and by printing multiple times.

The antifungal efficacy of terbinafine HCl, used as a model drug, was tested using disc diffusion assay with drug printed onto antimicrobial discs and on human nail clippings. Both the discs and human nail clipping diffusion assay showed inhibited growth of *T. rubrum*, compared to drug-free controls. Hence, the drug was able to penetrate through the nail clipping and into the agar plate. Both antifungal assays used low doses of terbinafine HCl, so the drug load can be increased to further enhance the fungal inhibition. Further work is required to fully optimise this system. In the future, this could be adapted to include an occlusive layer or multiple drugs for improved efficacy. There is also the potential to incorporate a system for automatic detection of the diseased part of the nail. Overall, inkjet printing presents a viable and exciting new method for dosing drugs onto nails. In particular, inkjet nail printing presents a feasible and favourable novel approach for the topical treatment of onychomycosis.

## Figures and Tables

**Figure 1 pharmaceutics-14-00448-f001:**
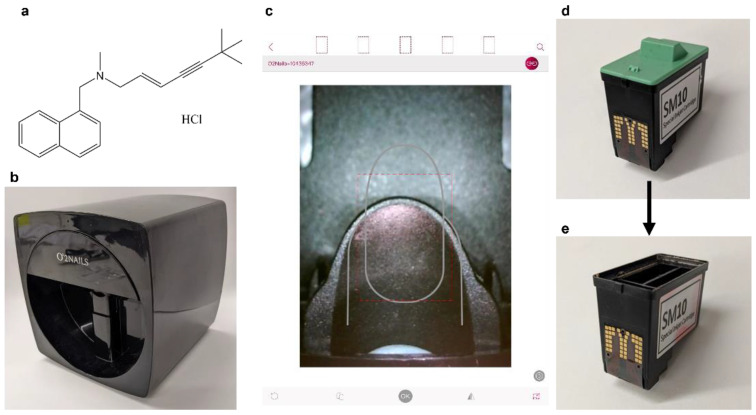
(**a**) Chemical structure of terbinafine HCl, the active pharmaceutical ingredient. (**b**–**e**) Images of the O2Nail V11 nail printer. (**b**) Picture of the nail printer. (**c**) Image of the nail printing app, taken from the iPad screen. The display is from the in-printer camera. (**d**) Picture of the SM10 Inkjet cartridge, with the lid on. (**e**) Picture of the ink cartridge, after removing the lid and cleaning out the commercial inks. This cartridge can then be loaded with an in-house prepared solution for printing.

**Figure 2 pharmaceutics-14-00448-f002:**
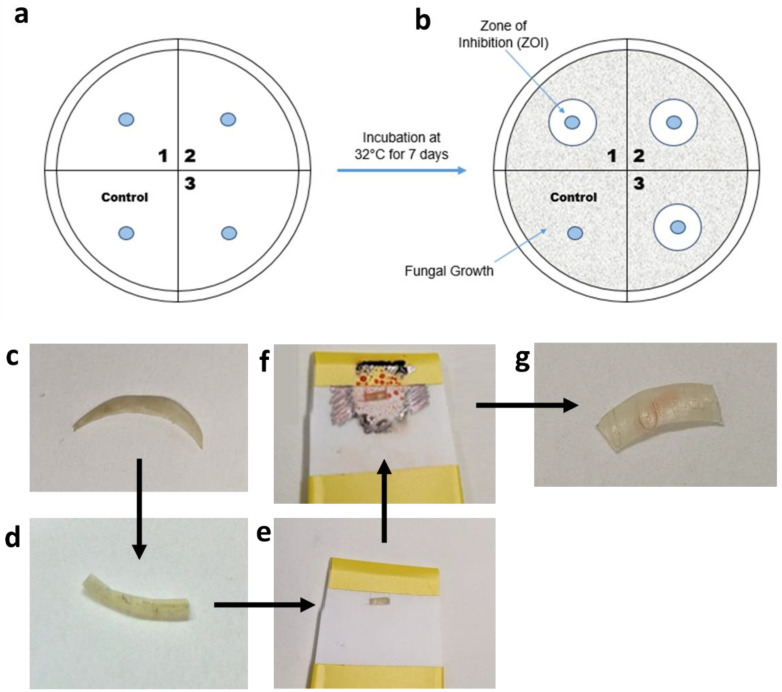
(**a**,**b**) Diagrams of the disc diffusion assay procedure. (**c**–**g**) Pictures of the steps taken in nail printing. (**c**) The initial nail is chosen, and is cut in half (**d**) and then loaded into the nail holder (**e**). (**f**) The nail holder is printed onto and, once dried, the nail can be removed from the printer (**g**).

**Figure 3 pharmaceutics-14-00448-f003:**
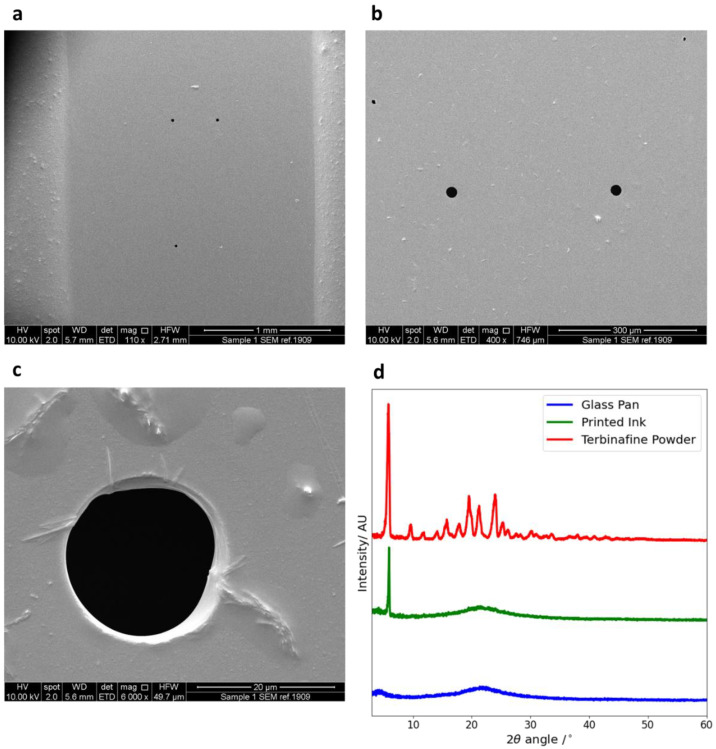
(**a**–**c**) SEM image taken of the nozzles in order to determine the nozzle diameter. The magnifications are 110× (**a**), 400× (**b**) and 6000× (**c**). Taken across the values, the diameter of the orifice was measured as 21.0 ± 2.2 µm. (**d**) The XRD data for the glass pan, printed ink and terbinafine HCl powder.

**Figure 4 pharmaceutics-14-00448-f004:**
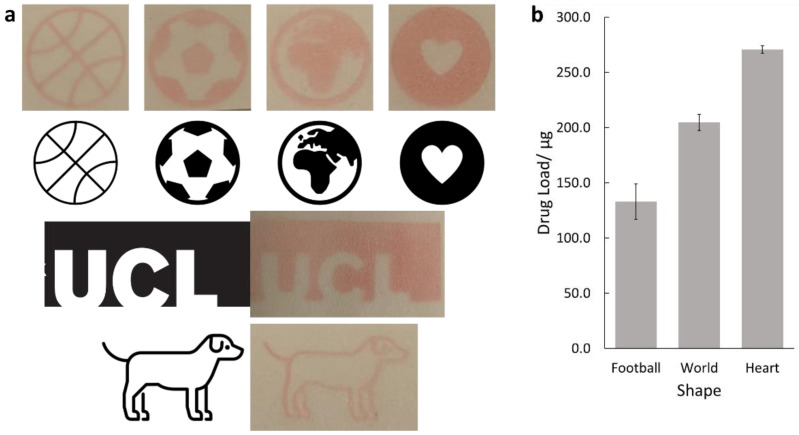
(**a**) Images of the selected black and white shapes and photographs of the printed shapes. The images selected were a basketball, a football, a world, a heart, the University College London logo and a cartoon dog. (**b**) The drug load for the football, world and heart shapes. All three show high levels of drug loading, with the heart showing the highest.

**Figure 5 pharmaceutics-14-00448-f005:**
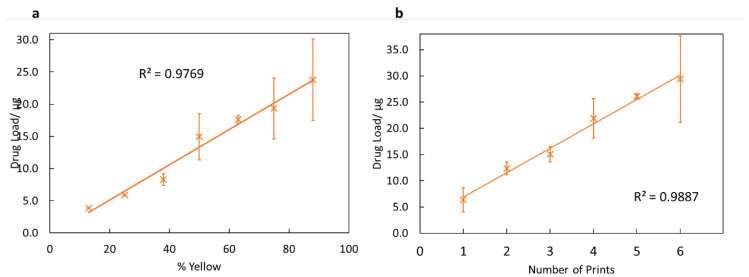
Plots demonstrating methods for controlling the drug dose applied. (**a**) A graph showing the results for how the drug load changes with the changing percentage of yellow of the object. The percentage of yellow corresponds with the lightness of the yellow. Error bars show the standard deviations. (**b**) A graph showing how the number of times printing was conducted affects the total drug load. Error bars show the standard deviations.

**Figure 6 pharmaceutics-14-00448-f006:**
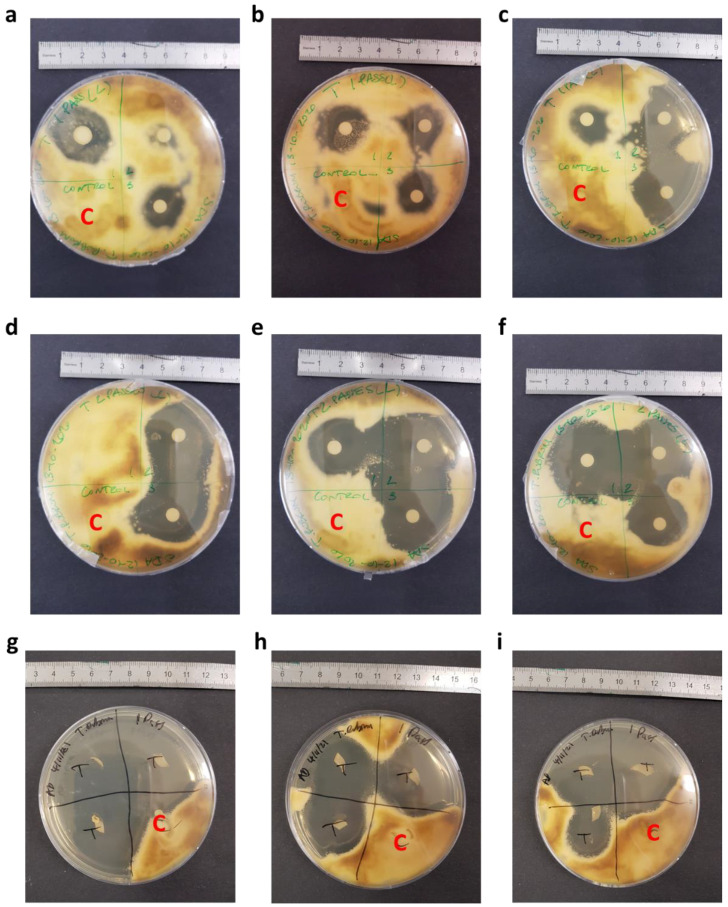
Photographic images of the assay plates, demonstrating the inhibition of *T. rubrum* by both printed-on discs and printed-on nails. (**a**–**c**) Disc diffusion assay for 1 pass of terbinafine HCl at 29 µg/mL. (**d**–**f**) Disc diffusion assay for 2 passes of terbinafine HCl at 29 µg/mL. (**g**–**i**) Nail diffusion assay for 1 pass of Terbinafine HCl at 5 mg/mL. The units on the rulers shown are cm, and the control quadrants are labelled with a red ‘C’. The quadrants labelled ‘1’, ‘2’, ‘3’ and ‘T’ are all test quadrants.

**Table 1 pharmaceutics-14-00448-t001:** Density, viscosity and surface tension values for various solutions. These values are critical for ensuring any inks developed will be printable with the two-dimensional inkjet printer.

Solution	Density (g cm^−3^)	Viscosity (mPa·s^−1^)	Surface Tension (mN·m^−1^)	Z Value
Yellow Commercial Ink	1.052	1.99	35.70	14.11
Magenta Commercial Ink	1.03	2.24	35.63	12.39
Cyan Commercial Ink	1.00	2.28	35.50	11.98
Ethanol and Water	0.828	1.97	26.33	10.86
Ethanol, Water and 1 mg/mL Terbinafine HCl	0.831	1.95	26.30	10.99
Ethanol, Water, 1 mg/mL Terbinafine HCl and Food Colouring	0.809	2.00	26.37	10.58

## Data Availability

Data can be made available upon reasonable request.

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
