# Peer review of "Printing Drugs onto Nails for Effective Treatment of Onychomycosis"

_pharmaceutics, 2022, doi:10.3390/pharmaceutics14020448_

Round 1
Reviewer 1 Report
The present manuscript (pharmaceutics-1583474) is nicely written and provides novel and interesting findings in the area of drug delivery. Indeed, following suggestions need to be considered for further improvement.
- Physicochemical characteristics of terbinafine hydrochloride should be discussed in the introduction. As per the physicochemical characteristics of drugs its compatibility/stability in printing ink should be rationalized in the study design.
- Stability of the drug (terbinafine hydrochloride) in ink should be measured and detailed analysis considering this perspective should be incorporated in the revised manuscript.
- Limitations of the current investigation and future perspectives should be incorporated in the conclusion section.
Author Response
|
Comment |
Reply |
|
Physicochemical characteristics of terbinafine hydrochloride should be discussed in the introduction. As per the physicochemical characteristics of drugs its compatibility/stability in printing ink should be rationalized in the study design |
We would like to thank the review for their comment. We have now included more details on the physicochemical characteristics of the drug (lines 54-58), as well as a reference to the thermal stability of different crystal structures of Terbinafine HCl (lines 58-59, 310-312). |
|
Stability of the drug (terbinafine hydrochloride) in ink should be measured and detailed analysis considering this perspective should be incorporated in the revised manuscript |
We would like to thank the reviewer for their feedback. We have now included some preliminary data for the stability of the drug in solution (lines 155-157, 318-323). However, we have not included detailed analysis, as this is only a proof-of-concept paper, and so we feel in-depth analysis would not be appropriate. |
|
Limitations of the current investigation and future perspectives should be incorporated in the conclusion section. |
We would like to thank the reviewer for their comment. We have expanded the conclusion to include the suggested details (lines 526-529). |

Reviewer 2 Report
The current manuscript provides an account of printing drugs onto nails for effective treatment of onychomycosis. The rationale and motivation of the study is not clear, and I did not find the study interesting from a personalized medicine point-of-view (this being the major aspect of the study). The difference in drug loading among various shapes is predictable and can be easily based on surface area and volume. Why would one need 3D printing on the nails to treat onychomycosis? There are several nail lacquer based systems that can be loaded with drugs and can used as and when required by the patient themselves - better patient compliance, larger coverage, and cost-efficient. If the authors check the fragile and irregular nature of the nails, the 3D printing may not work. The controls should essentially include painted nail as well.
Author Response
|
Comment |
Reply |
|
The rationale and motivation of the study is not clear, and I did not find the study interesting from a personalized medicine point-of-view (this being the major aspect of the study). |
We would like to thank the reviewer for their feedback. We have now added additional information on why we selected the drug (lines 54-56), as well as some of the issues with the current research (lines 69-70) and advantages of the system compared to other devices (lines 479-483). Ultimately, the paper is a proof-of-concept and so does not aim to demonstrate a fully finished product but instead present a novel approach. |
|
The difference in drug loading among various shapes is predictable and can be easily based on surface area and volume. |
We would like to thank the reviewer for their comment. However, we feel they may have mis-interpreted the work; the printer here is a two-dimensional inkjet printer, and so surface area: volume is not relevant. We have added additional text to make it clear the printer is two-dimensional (lines 71, 167, 336). While the difference in drug loading might be obvious, we included that experiment to demonstrate that the printer can indeed accurately reproduce the shapes, that this gives a predictable pattern for drug loading and that the drug loads are high. This is explained in the manuscript on lines 382-386. Further emphasis on why this is important has been added (lines 386-387). |
|
Why would one need 3D printing on the nails to treat onychomycosis? There are several nail lacquer based systems that can be loaded with drugs and can used as and when required by the patient themselves - better patient compliance, larger coverage, and cost-efficient. |
We would like to thank the reviewer for their feedback. However, again we would like to emphasise that this work is for a two-dimensional printing system and not a three-dimensional printing system (see above response). We have further expanded the advantages of our system with respect to the existing literature (lines 479-483) and the potential advantages for this system in the future (lines 489-493). |
|
If the authors check the fragile and irregular nature of the nails, the 3D printing may not work. |
We would like to thank the reviewer for their comment. Since the total mass dispensed with each print is low, the fragility of the nail should not be an issue in this work. The irregular nature of the nails was tested in the work, with the results shown (Figure 6g-i). We have also added an additional comment on the irregular nature of the nails to the text (lines 463-465). |
|
The controls should essentially include painted nail as well. |
We would like to that the reviewer for this suggestion. In this work, the control nails were nail with drug-free solution printed onto them. This is the most applicable control in this case, as it eliminates any error from inhibition due to the printing process. The use of commercial nail polish would have been inappropriate, as these have been demonstrated in the literature to indirectly contribute to the transmission of onychomycosis [1,2] and would not have shown that the drug is the cause of inhibition. |

Reviewer 3 Report
The authors of the manuscript "Printing drugs onto nails for effective treatment of onychomycosis" evaluate the treatment of onychomycosis utilizing inkjet printing and, specifically, in-house prepared drug-loaded inks. Terbinafine hydrochloride was used as model drug. The physico-chemical characterization (i.e., density, viscosity, surface tension) of the commercial and in-house prepared drug inks took place. Furthermore, the printed drug dose and the in vitro antifungal efficacy of the formulations were evaluated. Some comments that will improve the manuscript’s quality are summarized below.
- The introduction is rather short and therefore the authors are recommended to describe more thoroughly the effects of onychomycosis and the subsequent challenges on treatment, in order to highlight the significance of the examined treatment technology.
- In line 74, the API is referred as terbinafine hydrochloride, in line 142 terbinafine HCL and in line 305 the authors mentioned just “terbinafine”. Please use the exact form and abbreviation.
- The authors are also recommended to include the chemical structures of the API.
- The authors should provide some information about the commercial inks that used for the comparison with the in-house prepared.
- In line 322, the authors claimed that the physical state of the API “does not appear to be significantly altered”, however, there are significant differences between the obtained XRD diffractograms. Please revert.
- Please check line 334.
- Lines 366-368 and 372-373 contain a very bold statement without further explanations. Please discuss the possible mechanisms.
- The authors could also improve the discussion by highlighting the advantages of the proposed delivery system with respect to similar ones. In fact, an interpretation of the results has been provided in Section 3 but only minimal referencing to what has been already presented in the literature is made.
Author Response
|
Comments |
Reply |
|
The introduction is rather short and therefore the authors are recommended to describe more thoroughly the effects of onychomycosis and the subsequent challenges on treatment, in order to highlight the significance of the examined treatment technology. |
We would like to thank the reviewer for this comment. We have now added additional information on some of the symptoms of onychomycosis (lines 51-53), as well as a more expanded discussion on terbinafine (lines 54-59). |
|
In line 74, the API is referred as terbinafine hydrochloride, in line 142 terbinafine HCL and in line 305 the authors mentioned just “terbinafine”. Please use the exact form and abbreviation. |
We would like to thank the reviewer for bringing this to our attention. We have now edited the manuscript so that it is clear that terbinafine HCl was used throughout, and that this is the same as terbinafine hydrochloride. |
|
The authors are also recommended to include the chemical structures of the API |
We would like to thank the reviewer for this suggestion. The API chemical structure has now been added as Figure 1a, with the structure referenced in line 85. |
|
The authors should provide some information about the commercial inks that used for the comparison with the in-house prepared |
We would like to thank you for this suggestion. However, as the composition of the commercial inks is proprietary, no additional information other than the results of the tests completed in this work is available. We have modified the manuscript to make it explicit that the ink composition is proprietary information (lines 170) |
|
In line 322, the authors claimed that the physical state of the API “does not appear to be significantly altered”, however, there are significant differences between the obtained XRD diffractograms. Please revert. |
We would like to thank the reviewer for bringing this to our attention. The manuscript has been edited (lines 349-353) to reflect that the drug is in fact partially crystalline. |
|
Please check line 334 |
Thank you for bringing this to our attention; the error has been removed. |
|
Lines 366-368 and 372-373 contain a very bold statement without further explanations. Please discuss the possible mechanisms |
We would like to thank the reviewer for this suggestion; the manuscript has been altered (lines 395-400, 404-405) to make the mechanisms and explanations more explicit |
|
The authors could also improve the discussion by highlighting the advantages of the proposed delivery system with respect to similar ones. In fact, an interpretation of the results has been provided in Section 3 but only minimal referencing to what has been already presented in the literature is made |
We would like to thank the review for this suggestion and insight. To the best of our knowledge, this is the first manuscript to demonstrate inkjet printing of terbinafine and the first to establish inkjet printing onto nails as a method of drug delivery, and so the number of similar publications is sparse. However, additional comparisons on the advantages of this system have been added throughout the manuscript (lines 432-433, 479-483) |

Round 2
Reviewer 2 Report
No further comments.
Reviewer 3 Report
All comments and recommendations were appropriately addressed by the authors. The paper is now compliant with the Pharmaceutics’ standards and, in my opinion, is suitable for publication.